# Healthy Lifestyle and Incidence of Metabolic Syndrome in the SUN Cohort

**DOI:** 10.3390/nu11010065

**Published:** 2018-12-30

**Authors:** Maria Garralda-Del-Villar, Silvia Carlos-Chillerón, Jesus Diaz-Gutierrez, Miguel Ruiz-Canela, Alfredo Gea, Miguel Angel Martínez-González, Maira Bes-Rastrollo, Liz Ruiz-Estigarribia, Stefanos N. Kales, Alejandro Fernández-Montero

**Affiliations:** 1Department of Occupational Medicine, University of Navarra, 31008 Pamplona, Navarra, Spain; mgarralda.7@alumni.unav.es; 2Department of Preventive Medicine and Public Health, University of Navarra, 31008 Pamplona, Navarra, Spain; scarlos@unav.es (S.C.-C.); jdiaz.14@alumni.unav.es (J.D.-G.); mcanela@unav.es (M.R.-C.); ageas@unav.es (A.G.); mamartinez@unav.es (M.A.M.-G.); mbes@unav.es (M.B.-R.); lruiz.29@alumni.unav.es (L.R.-E.); 3IDISNA, Navarra Health Research Institute, 31008 Pamplona, Navarra, Spain; 4CIBER Fisiopatología de la Obesidad y Nutrición (CIBER Obn), Instituto de Salud Carlos III, 28029 Madrid, Spain; 5Department of Nutrition, Harvard T.H. Chan School of Public Health, Boston, MA 20115, USA; 6Department of Environmental Health, Harvard T.H. Chan School of Public Health, Boston, MA 20115, USA; skales@hsph.harvard.edu

**Keywords:** healthy lifestyle score, metabolic syndrome, SUN cohort

## Abstract

We assessed the relationship between a healthy lifestyle and the subsequent risk of developing metabolic syndrome. The “Seguimiento Universidad de Navarra” (SUN) Project is a prospective cohort study, focused on nutrition, lifestyle, and chronic diseases. Participants (*n* = 10,807, mean age 37 years, 67% women) initially free of metabolic syndrome were followed prospectively for a minimum of 6 years. To evaluate healthy lifestyle, nine habits were used to derive a Healthy Lifestyle Score (HLS): Never smoking, moderate to high physical activity (>20 MET-h/week), Mediterranean diet (≥4/8 adherence points), moderate alcohol consumption (women, 0.1–5.0 g/day; men, 0.1–10.0 g/day), low television exposure (<2 h/day), no binge drinking (≤5 alcoholic drinks at any time), taking a short afternoon nap (<30 min/day), meeting up with friends >1 h/day, and working at least 40 h/week. Metabolic syndrome was defined according to the harmonizing definition. The association between the baseline HLS and metabolic syndrome at follow-up was assessed with multivariable-adjusted logistic regressions. During follow-up, we observed 458 (4.24%) new cases of metabolic syndrome. Participants in the highest category of HLS adherence (7–9 points) enjoyed a significantly reduced risk of developing metabolic syndrome compared to those in the lowest category (0–3 points) (adjusted odds ratio (OR) = 0.66, 95% confidence interval (CI) = 0.47–0.93). Higher adherence to the Healthy Lifestyle Score was associated with a lower risk of developing metabolic syndrome. The HLS may be a simple metabolic health promotion tool.

## 1. Introduction

Metabolic syndrome (MetSyn) is characterized by the clustering of several metabolic abnormalities frequently observed in clinical practice: Abdominal obesity, dyslipidemia, hyperinsulinemia, impaired fasting glucose, and high blood pressure, according to the International Diabetes Federation, the American Heart Association, and the National Heart, Lung, and Blood Institute harmonizing definition [1]. In a prospective cohort study done on middle-aged healthy men, MetSyn was associated with cardiovascular disease and mortality, and a published meta-analysis of longitudinal studies revealed an association between MetSyn and a higher risk of developing type 2 diabetes mellitus, cardiovascular disease, atherosclerosis, and higher all-cause mortality [2,3]. Due to the existence of different definitions for diagnosing this syndrome, prevalence estimates vary. However, it is accepted that the prevalence of MetSyn generally increases as body mass index and age increase [4]. In developed countries, the prevalence of MetSyn is about 25% of the adult population [5,6,7], and its incidence has been increasing over the last years. In Spain, MetSyn prevalence reached 10% in 2005 [8], and it increased to over 30% by 2012 [9]. Genetics cannot explain these differences alone, and environmental influences play an important role.

Several articles have shown the association between different lifestyle habits and the risk of developing MetSyn according to the harmonizing definition. In this context, eating habits are considered modifiable determinants of MetSyn. Prospective studies (that included participants who were young–middle-aged adults and that considered the harmonizing definition of MetSyn) and a systematic review (that evaluated studies with adults >18 years and that considered the ATPIII definition) analyzing nut consumption, sweet beverages, or adherence to Mediterranean diet patterns, as well as a meta-analysis on this topic (that worked with trials including adults >29 years and used the ATPIII criteria) have proven this association [10,11,12,13]. In a prospective study from the “Seguimiento Universidad de Navarra” (SUN) (that included young–middle-aged adults and used the harmonizing definition of MetSyn), physical activity was significantly associated with a lower risk of developing MetSyn [14], whereas, according to a meta-analysis (that analyzed studies whose participants were young–middle-aged adults and that used the MetSyn criteria proposed by the WHO, ATPIII, modified ATPIII, and ACE/AACE) and a longitudinal population-based study (that analyzed 43-year-old adults and used the IDF definition), active smoking and time spent viewing TV were associated with higher risks of MetSyn [15,16]. Several studies evaluating middle-aged adults and using the AHA definition, as well as studies in people who were overweight or obese, have evaluated the effect of the combination of classical healthy life factors on MetSyn (smoking, drinking, dietary habits, and physical activity) with more complex nutritional indexes and have reported them to be significantly associated with MetSyn [17,18]. These previous articles worked with middle-aged adults.

In conclusion, classic healthy lifestyle habits have proven to reduce the risk of developing MetSyn.

In a recent longitudinal study conducted in 2017 in the SUN cohort [19] (in young–middle-aged adults), a new Healthy Lifestyle Score (HLS) was associated with a significant reduction in cardiovascular disease (CVD). This HLS basically included the traditional cardiovascular healthy lifestyle factors (i.e., tobacco, alcohol, diet, and physical activity), and it also took into account other modern life habits including time spent watching television, binge drinking, napping, social life with friends, and number of hours spent working.

Therefore, since it is well known that the MetSyn is a risk factor for CVD, an important question from a public health perspective is whether a healthy lifestyle would also reduce the risk of MetSyn. Our hypothesis was that a higher adherence to the HLS would be associated with a lower risk of MetSyn.

The aim of this study was to prospectively analyze the effectiveness of this new and easy-to-apply HLS in the reduction of MetSyn risk.

## 2. Materials and Methods

### 2.1. Study Design

The “Seguimiento Universidad de Navarra” (University of Navarra Follow-Up) Project is a dynamic prospective cohort study that has been conducted in Spain since December 1999 with permanently open recruitment of university graduates. It was designed based on the model of other large cohort studies conducted at the Harvard School of Public Health (the Nurses’ Health Study and the Health Professionals Follow-Up Study). Additional details on its objectives, design, and methods have been previously published [20].

Information is mainly gathered through self-reported questionnaires. Participants’ information is collected biennially through mailed or electronically mailed questionnaires. Upon completion of the first questionnaire (Q_0), including a total of 554 items used as baseline information, participants receive, every other year, different follow-up questionnaires. These contain important questions to evaluate changes in lifestyle and health-related behaviors, anthropometric measures, incident diseases, and medical conditions. Participants of all ages may be included in the SUN cohort, but they must have had university studies. This inclusion criteria allows for a better control of confounding by education-related variables and by making the interpretation of results easier and therefore adding internal validity to the high-quality information derived from the questionnaires.

### 2.2. Participants

In our study, a subsample of the SUN cohort was selected. In order to achieve a minimum follow-up of 6 years, only participants who had at least completed the 6-year follow-up questionnaire (Q_6) were included. There were 20,622 participants eligible to be included (SUN participants who responded to the baseline questionnaire (Q_0) before March 2010). We excluded 5080 participants who had either prevalent MetSyn or any MetSyn component at baseline. We also excluded 1399 participants who had extremely low or high total energy intake [21], as well as 1411 participants lost to follow-up (retention rate = 92.7%) and 1985 subjects who did not provide all of the relevant information to diagnose metabolic syndrome at the 6-year follow-up (Figure 1). Therefore, there were 10,807 participants available for analysis. 

Informed consent was obtained from all individual participants included in the study by the voluntary completion of the baseline questionnaire once participants understood the specific information needed, the methods used to deliver their data, and the future feedback from the research team. We asked their permission before any follow-up on their medical history. We informed the potential candidates of their right to refuse to participate in the SUN study or to withdraw their consent to participate at any time without reprisal, according to the principles of the Declaration of Helsinki. The Institutional Review Board of the University of Navarra approved these methods.

### 2.3. Exposure Assessment: Healthy Lifestyle Score Variables

We gathered information from the baseline questionnaires, which collected data on sociodemographic, clinical, anthropometric variables, and lifestyle aspects. Various studies have validated data from the self-reported questionnaires in the SUN cohort: Both anthropometric [22] and physical activity [23] data were analyzed in cohort subgroups. We used the validated 136-question semiquantitative food frequency questionnaire (FFQ) [24] for the evaluation of Mediterranean diet adherence, which was estimated with the Trichopoulou score (0–8 points) [25], (alcohol was excluded). So as to collect information on alcohol consumption, data was obtained through this questionnaire and other additional items related to alcohol consumption in the baseline questionnaire.

In order to assess adherence to a healthy lifestyle, 9 habits of the Healthy Lifestyle Score [19] were used (Table 1), excluding body mass index (BMI), as it is strongly related to MetSyn. The information for this score was gathered from the baseline questionnaire. Each participant received one point for each of the following 9 habits: Never smoking, moderate to high physical activity (>20 MET-h/week), Mediterranean diet (≥4 adherence points), moderate alcohol consumption (women, 0.1–5.0 g/day; men, 0.1–10.0 g/day; abstainers excluded), low television exposure (<2 h/day), no binge drinking (≤5 alcoholic drinks at any time), taking a short afternoon nap (<30 min/day), meeting up with friends >1 h/day, and working at least 40 h/week. This HLS could range from 0 (worst lifestyle) and 9 points (best lifestyle).

### 2.4. Outcome Assessment: Metabolic Syndrome and Assessment of Other Variables

The study outcome was incidence of MetSyn. We followed the harmonized definition of MetSyn according to the International Diabetes Federation; the National Heart, Lung, and Blood Institute; the American Heart Association; the World Heart Federation; the International Atherosclerosis Society; and the International Association for the Study of Obesity. According to this definition, metabolic syndrome consists of at least three abnormal findings out of the following 5 criteria [1]: (i) Central adiposity (≥94 cm for men and ≥80 cm for women, cut-off points for European populations); (ii) elevated triglycerides (TAG) (≥150 mg/dL or presence of pharmacological treatment for hypertriglyceridemia); (iii) reduced high-density lipoprotein cholesterol (HDL-cholesterol) (<40 mg/dL for men and <50 mg/dL for women or presence of pharmacological treatment for reduced HDL-cholesterol); (iv) elevated blood pressure (systolic ≥130 mmHg or diastolic ≥85 mmHg or presence of pharmacological treatment for hypertension in patients with a history of this disease); and (v) fasting glucose metabolism (≥100 mg/dL or pharmacological treatment for hyperglycemia).

To obtain the clinical criteria needed for the diagnosis of MetSyn, we used self-reported information provided by participants during the follow-up questionnaires. In the 6-year and 8-year follow-up questionnaires (Q_6 and Q_8), self-reported data about these specific MetSyn criteria were collected. All participants were sent a measuring tape with the Q_6 and Q_8 follow-up questionnaires, together with an explanation on how to measure their waist circumference by using the horizontal plane, midway between the inferior margin of the ribs and the superior border of the iliac crest [26]. Accuracy and validation of all self-reported data on MetSyn components had been previously analyzed in a specific subsample study from the SUN Project of 287 participants [27]. All the analytical parameters used in the validation of self-reported metabolic syndrome components were obtained from the Clinical Analyses Service of Clínica Universidad de Navarra (CUN). The analyses of glucose, HDL-cholesterol, total cholesterol, and triglycerides were measured in blood serum with the analyzer equipment Roche/Hitachi Modular Analytics, and through spectrophotometry by the enzymatic colorimetric method with glucose oxidase and p-aminophenazone (GOD-PAP). High intraclass correlations were found for waist circumference (*r* = 0.86, 95% confidence interval (CI): 0.80–0.90) and triglycerides (*r* = 0.71, 95% CI: 0.61–0.79), whereas moderate intraclass correlations were found (between 0.46 and 0.63) for the other factors. An additional study, which compared the validity of self-reported diagnosed MetSyn and MetSyn diagnosed by the medical records of the participants, was conducted in another subsample of the SUN Project [28]. Using ATP III criteria, 91.2% of MetSyn and 92.2% (95% CI: 85.7–96.4) of non-MetSyn cases were confirmed.

An incident case of MetSyn was defined when a participant, free of this condition or any of its components at baseline, met three or more of the criteria after at least 6 years of follow-up.

### 2.5. Ascertainment of Covariates

The baseline questionnaire also collected information of potential confounding factors between HLS and MetSyn such as sociodemographic characteristics (sex, age, and education level), sleep, medical history (prevalence of cancer, cardiovascular disease, and depression), dietary factors (following a special diet and total energy intake (Kcals/day)), and anthropometric data (BMI). Our approach was to use the consideration of a priori causal knowledge to suggest which were the most relevant variables to be adjusted for. Causal diagrams were used to encode qualitative a priori subject matter knowledge. We did not use merely statistical criteria, because this statistics-only approach has been discouraged [29,30,31]. For instance, in reference to the prevalence of cancer, it may lead to weight loss and therefore be associated with MetSyn. Besides, it can be related to a change in diet and lifestyle, and these can influence MetSyn.

### 2.6. Statistical Analyses

According to their baseline HLS, participants were classified into 5 groups to ensure an appropriate sample distribution with sufficient participants and incident cases in each category. Thus, we merged extreme categories, and the distribution of these five categories was 0–3, 4, 5, 6, and 7–9 points. Logistic regression models were fit to assess the risk of metabolic syndrome (MetSyn) after a 6-year follow-up according to HLS categories. Odds ratios (ORs) and their 95% confidence intervals (95% CIs) were calculated considering the lowest category (0–3) as the reference. Linear trend tests were calculated by assigning the median score of each category to all participants in that category and treating this variable as a continuous variable.

For all the analyses, we fitted a crude model, an age- and sex-adjusted model, and a multivariable adjusted model using the following covariates as confounding factors: Age, sex, depression (yes/no), education level (technical/nongraduated, graduated, postgraduate, master’s, doctorate), cardiovascular disease (yes/no), prevalent cancer (yes/no), following any special diet (yes/no), body mass index (kg/m^2^), energy intake (kcal/day), hours of sleep (h/day), and year of questionnaire completion.

Additional multivariable adjusted analyses were conducted to test the association between the HLS categories and each of the individual criteria for MetSyn.

To assess the individual contribution of each specific factor of the HLS score to the risk of MetSyn, logistic regression models were fitted for each of the nine indicators of healthy life habits, adjusting for the effect of the rest of the elements that constituted the index. The reference category was the absence of the habit of healthy life (score 0 on the specific element).

We used the imputation approach because we had some missing information in important variables such as time spent watching TV, having a short afternoon nap, and time spent with friends. This statistical technique tries to overcome the problem that single imputed values are not actually observed but predicted values, and attributes the most probable value [32]. To carry out the imputation approach, we took into account potential confounding factors, each of the other components of Healthy Lifestyle Score, and each one of the components of MetSyn.

Sensitivity analyses were performed to ensure the robustness of the results in different scenarios. We repeated the analyses stratifying by age (≥45) and sex and without imputation of the lost variables.

All *p*-values presented are two-tailed, and *p* < 0.05 was considered to be statistically significant. Analyses were performed using STATA/SE version 12.0.

## 3. Results

The main characteristics of participants according to the HLS categories are shown in Table 2. Compared to subjects who had lower HLSes (0–3 points), those who had the highest score (7–9 points) were less likely to be women, consumed less alcohol per day, had less prevalent depression, were more likely to follow a special diet, and had a slightly higher total energy intake per day. There were no differences in age, baseline BMI, and the prevalence of CVD and cancer.

Of the participants, 458 (4.24%) (272 men, 186 women) initially free of metabolic syndrome (MetSyn) were newly diagnosed as incident cases during the 6-year follow-up. Those who had high HLSes (7 to 9 points) had a significant 34% lower risk of developing MetSyn than those who had lower HLSes (0 to 3), after adjusting for other factors related to MetSyn (Table 3).

Table 4 shows the multivariable-adjusted ORs for each component of MetSyn across HLS categories after the 6-year follow-up. With the exception of HDL-cholesterol, all point estimates of ORs for the upper versus lower category of the HLS suggested inverse associations. However, only the associations with waist circumference and elevated blood pressure showed statistically significant associations (*p* for trend < 0.05).

Figure 2 shows the multivariable-adjusted ORs across the 9 habits of the HLS and the risk of MetSyn. Only low television exposure (<2 h/day) and a short afternoon nap (<30 min/day) were significantly related to the incidence of MetSyn.

In the stratified analysis, we found a significant inverse association between HLS and MetSyn in men and in those older than 55 years, but we did not find any significant interaction (Figure 3). When we performed the analyses without imputation, the results did not change in magnitude, but they were no longer significant (OR = 0.67, 95% CI = 0.44–1.02).

## 4. Discussion

This prospective study of initially healthy young–middle-aged Mediterranean university graduates showed that a high adherence to a Healthy Lifestyle Score (7 to 9 points) was associated with a lower risk of incident MetSyn after 6 years of follow-up compared to participants with the lowest number of healthy lifestyle factors (0 to 3). Regarding the components of MetSyn, the highest inverse association of this HLS was observed for a high waist circumference and blood pressure. Although we observed a strong and consistent association, only two healthy lifestyle factors, reduced time watching TV and napping less than 30 min, were significantly associated with a risk reduction of MetSyn. This finding suggests that the synergistic effect of the combination of several lifestyle factors is probably more important than the individual effect of each one of them when considered individually.

The HLS has already been shown to present a strong inverse association with cardiovascular disease in this same cohort [19]. Our study is of interest given that MetSyn is not only a risk factor for cardiovascular diseases, but also for type 2 diabetes, atherosclerosis, and all-cause mortality [17,18], among others.

Our findings were consistent with several previous studies supporting the beneficial impact of combinations of healthy lifestyle behaviors on the primary prevention of MetSyn. These studies demonstrated that the risk of MetSyn decreased as the number of lifestyle factors increased, suggesting that a constellation of factors rather than a single factor is more associated with decreased risks of MetSyn [17,18,33,34].

Our findings were also consistent with previous studies demonstrating an association between lifestyle scores and health-related outcomes, such as mortality [35] and cardiovascular diseases [19]. A recent prospective cohort study conducted in the United States showed that five healthy lifestyle-related factors—physical activity (>30 min/day of moderate or vigorous activities), a healthy diet, moderate alcohol consumption, never smoking, and a normal BMI (18.5 to 24.9 kg/m^2^)—are associated with lower risk of all-cause mortality (hazard ratio (HR) 0.26 (95% CI, 0.22–0.31)), cancer mortality (HR 0.35 (95% CI, 0.27–0.45)), and CVD mortality (HR 0.18 (95% CI, 0.12–0.26)) compared to participants with zero low-risk factors [36]. Similar findings of reduced mortality were demonstrated in another prospective cohort study of Spanish older adults, investigating the combined impact between three traditional (diet, physical activity, and smoking) and three nontraditional health behaviors (social interaction, sedentary time, and sleep duration) on mortality [37].

Our HLS combined indicators of lifestyle habits (never smoking, physical activity, Mediterranean diet, and moderate alcohol consumption) with other factors not typically included in risk scores (television exposure <2 h/day, no binge drinking, taking a short afternoon nap (<30 min), meeting up with friends for more than 1 h/day, and working at least 40 h/week).

According to the literature, and consistent with our results, smoking is associated with a higher risk of developing MetSyn [15,38,39]. Physical activity is also associated with a reduction in the risk of MetSyn [14,40,41,42]: However, we did not find a significant association between physical activity and MetSyn, only a trend. 

Many studies have shown, in agreement with our results, the association between the Mediterranean dietary pattern and the risk of MetSyn [43,44,45,46]. However, this finding did not reach statistical significance. A potential explanation for these results may rely on the fact that our population was composed of healthy young–middle-aged Mediterranean university graduates, as shown in Table 2. In addition, the Mediterranean diet score proposed by Trichopoulou et al. [25,47] included alcohol intake: Nevertheless, in our study this was considered a separate lifestyle element because the literature suggests that excessive consumption is related to an increased risk of MetSyn [48,49]. Inconsistent with the literature, our results showed that avoidance of “binge drinking” was not associated with a lower risk of MetSyn. However, this could be explained by the fact that we defined avoidance of binge drinking as never having had more than five alcoholic drinks in a single occasion. Therefore, since the prevalence of binge drinking in Spain has been found to be moderately high [50,51], people with healthy lifestyle habits were included in the unhealthy allocation, potentially leading to a non-differential misclassification bias.

Consistent with our results, some studies have reported the relation between TV viewing and MetSyn [52], as TV viewing may displace physical activity. A meta-analysis [53] demonstrated a J-curve relation between nap time and the risk of MetSyn. This study proved that longer than 40 min/day napping was associated with an increased risk of MetSyn.

Social relationships have been suggested as a protective factor, as they are positively related to minutes of physical activity per week and days of physical activity per week. However, they have been positively associated with number of servings of wine per week and increased high-density lipoprotein cholesterol [54], which may explain why our results showed no protection related to social relationships (variable: Spending time with friends) and MetSyn.

Finally, even if O’Reilly and Rosato found that professionals or managers who worked more than 40 h/week had a lower risk of death [55], which may be explained by maintaining a healthy lifestyle [56], other articles have not found a significant association [57]. In our study, working ≥40 h/week was associated with a higher risk of metabolic syndrome (OR 1.26; 95% CI 1.02–1.57): However, the results should be examined carefully, since our population consisted of university graduates whose work may be related to sedentary, seated jobs.

As expected, and consistent with the literature, adherence to various lifestyle habits showed a greater synergistic effect than a single habit in particular. Therefore, if individuals are concerned about their health, the total number of healthy habits should be increased [58].

It is important when interpreting our results to emphasize that the variables that made up this HLS were categorized in a dichotomous way. Most studies divide these variables into more categories to find greater differences between the extremes. This may be the reason why the differences found between healthy lifestyle factors analyzed individually (regarding the risk of MetSyn (Figure 2)) were more modest than in other investigations [38,44].

Therefore, healthy lifestyle scores can potentially be used as health promotion tools, which help people make health-conscious decisions regarding their behaviors. Future research should be conducted in different scenarios to better analyze the potential effects of HLSes on healthy behaviors and health outcomes.

The present study had several limitations. We observed an incidence of 4.24% of MetSyn. This incidence was lower than that described in the general population [8], but expected in a cohort of young adults with low baseline body mass index, a high educational level and, especially, after selecting only participants without any criteria for MetSyn at the baseline. As is the case in most cohort studies, the sample was not representative of the total population, and generalizing the results should be interpreted carefully. Another potential limitation was the self-reported data collection. Nevertheless, previously published validation studies were carried out in the SUN study, which evaluated the validity of our methods and the quality of the self-reported data by our highly trained volunteers. In any case, this would be expected to be nondifferential and would make the bias more likely to tend to null. Moreover, our analyses assumed that baseline habits remained stable throughout the 6-year follow-up, yet there might have been some changes, which would probably have led to underestimating the protective effects of the HLS. On top of this, as participants were young–middle-aged graduates and had few risk factors, there were few incidence cases. This could be associated with a lower statistical power. However, despite missing values of lifestyle behaviors being imputed, the magnitude of the results hardly changed.

The strengths of the present study included its dynamic participation, prospective design, long follow-up period, and high retention rate. As it is an open cohort, the number of participants is large and constantly increasing, which leads to more powerful results. Finally, validation studies were available for some variables, the outcomes were confirmed using medical records, and the findings were adjusted for a large number of covariables, therefore reducing the existence of potential confounding bias, although we cannot rule out the existence of residual confounding.

## 5. Conclusions

In summary, in this prospective cohort study of healthy young–middle-aged Mediterranean university graduates, a significant association was found between higher Healthy Lifestyle Scores and a reduction in the risk of incident metabolic syndrome. These results suggest the importance of promoting a comprehensive HLS to maintain metabolic health and allow for rapid evaluation in clinical practice. Further longitudinal and intervention studies in the general population should be conducted to confirm this relationship and to enable extrapolation.

## Figures and Tables

**Figure 1 nutrients-11-00065-f001:**
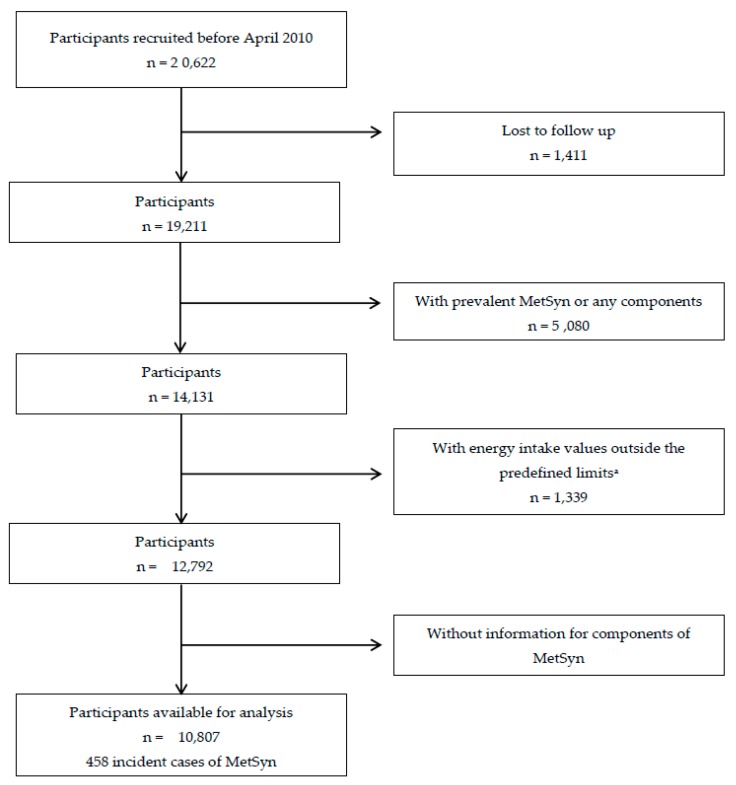
Flow chart depicting the selection process among participants of the Seguimiento Universidad de Navarra (SUN) cohort, 1999–2017; *n*: Number of participants; MetSyn: Metabolic Syndrome. ^a^ Total energy intake outside predefined limits (<800 or >4000 kcal/day for men, and<500 or >3500 kcal/day for women) [21].

**Figure 2 nutrients-11-00065-f002:**
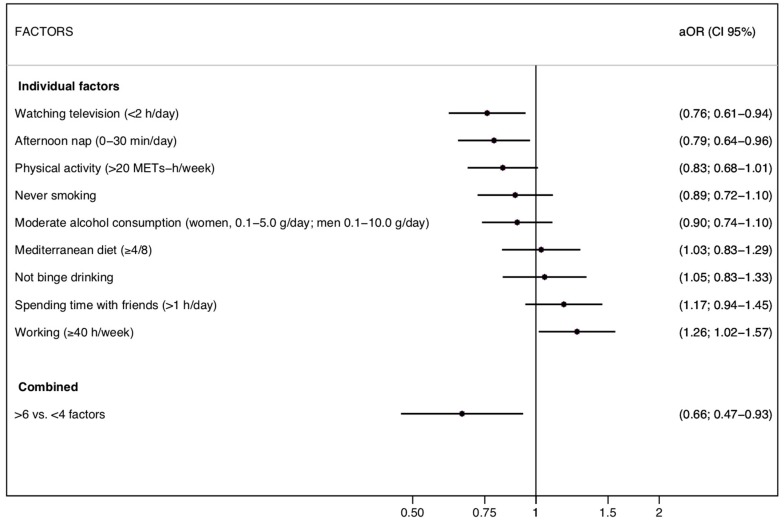
Risk of metabolic syndrome for each factor of the Healthy Lifestyle Score (the SUN cohort); aOR: Adjusted odds ratio; CI: Confidence interval.

**Figure 3 nutrients-11-00065-f003:**
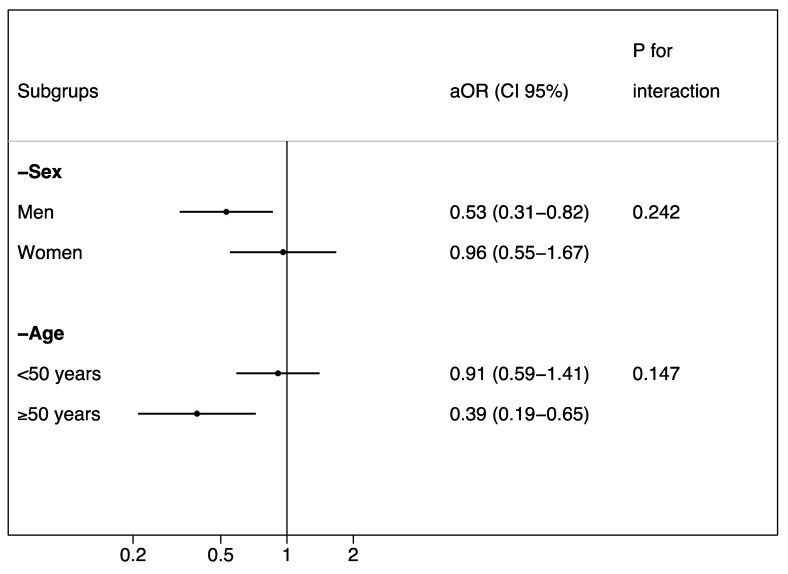
Risk of metabolic syndrome in the highest category compared to the lowest category of the Healthy Lifestyle Score. Stratified analyses (the SUN cohort); aOR: Adjusted odds ratio; CI: Confidence interval.

**Table 1 nutrients-11-00065-t001:** Healthy Lifestyle Score (HLS) ^a^.

	Score
Not smoking	
Never a smoker	1
Smoker (current and former)	0
Physical activity (MET-h/week)	
Physically active (>20 MET-h/week)	1
Not physically active (≤20 MET-h/week)	0
Mediterranean diet pattern (Trichopoulou score excluding alcohol) ^b^	
High adherence (≥4)	1
Low adherence (<4)	0
Moderate alcohol consumption	
Moderate consumption (women, 0.1–5.0 g/day; men, 0.1–10.0 g/day)	1
Abstention or high consumption (women >5 g/day; men >10 g/day)	0
Low time spent watching television	
Low television watching (<2 h/day)	1
High television watching (≥2 h/day)	0
Avoidance of binge drinking	
No binge drinking (≤5 alcoholic drinks at any time)	1
Binge drinking (>5 alcoholic drinks at any time)	0
Having a short afternoon nap	
Short afternoon nap (0.1–0.5 h/day)	1
Not having an afternoon nap or having a long nap (>0.5 h/day)	0
Time with friends	
Spending time with friends (>1 h/day)	1
Not spending time with friends (≤1 h/day)	0
Time working	
Full-time work (≥40 h/week)	1
Less than full-time work (<40 h/week)	0

^a^ Body mass index was excluded, as it is a component of metabolic syndrome. ^b^ Score from 0 to 8, because alcohol consumption was excluded.

**Table 2 nutrients-11-00065-t002:** Baseline characteristics of participants according to the number of Healthy Lifestyle Factors (HLFs) (the SUN cohort).

Number of Healthy Lifestyle Factors	0–3	4	5	6	7–9	*p*-Value
Participants, *n*	1468	1993	2599	2525	2222	
Sex, women (%)	69.8	68.0	68.3	66.3	63.7	<0.001
Age, years	35.8 ± 10.6	36.5 ± 11	35.9 ± 10.7	35.9 ± 10.8	34.2 ± 10	<0.001
Body mass index in men	24.9 ± 2.5	24.6 ± 2.3	24.6 ± 2.3	24.6 ± 2.2	24.3 ± 2.3	<0.001
Body mass index in women	22.0 ± 2.5	21.9 ± 2.5	21.9 ± 2.5	21.6 ± 2.4	21.6 ± 2.4	<0.001
Smoking, packs per year	8.8 ± 10.2	7.1 ± 9.4	5.4 ± 8.1	4.21 ± 7.8	1.9 ± 5.8	<0.001
Physical activity, MET-h/week	16.1 ± 15.2	21.4 ± 19.3	25.1 ± 21.8	30.3 ± 25	36.7 ± 26	<0.001
Mediterranean diet pattern ^a^	3.04 ± 1.54	3.56 ± 1.7	3.89 ± 1.73	4.24 ± 1.69	4.71 ± 1.49	<0.001
Alcohol consumption, g/day	8.0 ± 11.1	6.9 ± 9.4	6.2 ± 8.7	5.1 ± 7.1	3.9 ± 5.1	<0.001
Watching television, h/day	2.27 ± 1.51	1.81 ± 1.4	1.58 ± 1.31	1.33 ± 1.11	1.09 ± 0.83	<0.001
No binge drinking (%) ^b^	44.3	60.7	71	77.2	86.6	<0.001
Afternoon nap, min/day	0.3 ± 0.45	0.27 ± 0.39	0.25 ± 0.34	0.22 ± 0.29	0.22 ± 0.22	<0.001
Meeting up with friends, h/day	1.11 ± 1	1.24 ± 1.05	1.34 ± 1.09	1.39 ± 1	1.54 ± 1.03	<0.001
Working ≥40 h/week (%)	25.4	37.7	49.2	60.6	76.3	<0.001
Sleeping, h/day	7.5 ± 1.1	7.4 ± 1	7.4 ± 1	7.4 ± 0.9	7.4 ± 0.9	0.011
Depression disease (%)	12.7	11.2	10.7	9.2	9.3	0.001
Prevalent cardiovascular disease (%) ^c^	1.43	2.06	1.89	1.43	1.53	0.512
Prevalent cancer (%)	3.07	3.21	2.89	3.05	2.43	0.826
Education level						<0.001
No college (%)	8.86	9.13	8.96	10.18	10.26	
College (%)	26.6	22.9	25.2	23.1	21.1	
Postgraduate (%)	51.2	51.9	49.5	47.9	49.1	
Master’s (%)	5.93	7.02	7.7	8.08	8.51	
Doctorate (%)	7.43	9.03	8.66	10.73	10.98	
On any special diet (%)	4.36	6.77	5.73	6.81	7.25	0.003
Caloric consumption	2282 ± 607	2302 ± 594	2359 ± 603	2392 ± 606	2428 ± 593	<0.001

^a^ Trichopoulou score (from 0 to 8, with alcohol consumption excluded). ^b^ Less than 5 alcoholic drinks at any time. ^c^ Atrial fibrillation, paroxysmal tachycardia, coronFary artery bypass surgery or another revascularization procedure, heart failure, aortic aneurysm, pulmonary embolism, or peripheral venous thrombosis.

**Table 3 nutrients-11-00065-t003:** Incidence of metabolic syndrome at 6-year follow-up, according to the number of Healthy Lifestyle Score factors (the SUN cohort). OR: Odds ratio; CI: Confidence interval.

	Number of Healthy Lifestyle Factors					
	0–3	4	5	6	7–9	*p* for Trend
Participants, *n*	1468	1993	2599	2525	2222	
Incident cases	80	92	109	103	74	
Crude OR (95% CI)	1 (ref.)	0.86 (0.64–1.17)	0.73 (0.54–0.98)	0.75 (0.56–1.01)	0.57 (0.42–0.80)	0.002
OR adjusted for age and sex (95% CI)	1 (ref.)	0.77 (0.56–1.05)	0.68 (0.50–0.93)	0.69 (0.51–0.94)	0.60 (0.43–0.83)	0.003
Multivariable adjusted OR ^a^	1 (ref.)	0.82 (0.59–1.13)	0.72 (0.52–0.98)	0.76 (0.77–1.05)	0.66 (0.47–0.93)	0.027

ref: reference category. ^a^ Adjusted for age, sex, depression, education level, cardiovascular disease, prevalent cancer, following any special diet, body mass index, energy intake, hours of sleep, year of questionnaire completion.

**Table 4 nutrients-11-00065-t004:** Odds ratios and 95% confidence intervals for each component of metabolic syndrome at the 6-year follow-up, according to the number of HLFs (the SUN Cohort).

	Number of Healthy Lifestyle Factors					
	0–3	4	5	6	7–9	*p* for Trend
Waist Circumference (>94 cm men, 80 cm women) ^a^	1 (ref.)	0.88 (0.75–1.02)	0.74 (0.64–0.86)	0.74 (0.64–0.86)	0.68 (0.58–0.79)	<0.001
Elevated triglycerides (>150 mg/dL) ^a^	1 (ref.)	0.84 (0.64–1.12)	0.80 (0.62–1.07)	0.72 (0.54–0.95)	0.87 (0.66–1.16)	0.213
Reduced HDL-cholesterol (<40 mg/dL) ^a^	1 (ref.)	1.00 (0.73–1.38)	1.27 (0.95–1.70)	1.10 (0.81–1.49)	1.10 (0.80–1.51)	0.483
Elevated blood pressure (systolic >130 or diastolic >85 mmHg) ^a^	1 (ref.)	1.02 (0.85–1.24)	0.93 (0.78–1.12)	0.89 (0.74–1.07)	0.86 (0.71–1.05)	0.033
Elevated glucose (>100 mg/dL) ^a^	1 (ref.)	0.82 (0.64–1.05)	0.86 (0.68–1.09)	0.93 (0.74–1.17)	0.73 (0.57–0.94)	0.127

ref: reference category. ^a^ Adjusted for age, sex, depression, education level, cardiovascular disease, prevalent cancer, following any special diet, body mass index, energy intake, hours of sleep, year of questionnaire completion. HDL-cholesterol: High-density lipoprotein cholesterol.

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
