# Peer review of "Healthy Lifestyle and Incidence of Metabolic Syndrome in the SUN Cohort"

_nutrients, 2018, doi:10.3390/nu11010065_

Reviewer 1 Report

This manuscript is a gem of a study, a pleasure to read, and is a significant addition to the literature.

The topic is relevant, as a perusal of the literature about cardiometabolic disease will readily show. The study also offers some unique features, such as a social variable and nap time. The abstract and title adequately reflect the content and purpose.

The introduction describes the background, goals, and states the research question clearly.

The methods section describes what the reader needs to know about data collection, design, and statistics used.

The discussion is logical, chronological, comparative, and includes perspective and placement.

The authors demonstrate that they are sophisticated investigators but have a deep understanding of their topic. In particular, the limitations and strengths section is a fair representation that is unfortunately absent from many papers.

My suggestions here are very few> Figure 2 should have a much greater font size, and occupy a greater proportion of the page so it is easier to read.

On p7 line 217-218, the authors are over-conservative in claiming “biological plausibility.” The literature, as they know (and use in their bibliography) is rich with repeated confirmations of the associations of positive life style factors and healthy behaviors with subsequent cardiac, metabolic morbidities, mortality from cardiometabolic disease, and all-cause mortality. In some instances RCTs are available, and in some specific cases (eg triglyceride-rich lipoproteins), there is support from Mendelian randomization studies.

Pg 8 line 230—the authors are to be commended on this statement about HLS as potential health promotion tools. Perhaps another few sentences could be added (if they do not plan another publication about this application).

The discussion on p 8-9 is masterful.

The authors are to be congratulated on their fine work and contribution.

Author Response

Thanks for your review. Your comments help us and encourage us to improve our research.

We request your questions point by point below. The corrections suggested by you and other reviewers are highlighted in the original manuscript in red print.

This manuscript is a gem of a study, a pleasure to read, and is a significant addition to the literature.

The topic is relevant, as a perusal of the literature about cardiometabolic disease will readily show. The study also offers some unique features, such as a social variable and nap time. The abstract and title adequately reflect the content and purpose.

The introduction describes the background, goals, and states the research question clearly.

The methods section describes what the reader needs to know about data collection, design, and statistics used.

The discussion is logical, chronological, comparative, and includes perspective and placement.

The authors demonstrate that they are sophisticated investigators but have a deep understanding of their topic. In particular, the limitations and strengths section is a fair representation that is unfortunately absent from many papers.

My suggestions here are very few> Figure 2 should have a much greater font size, and occupy a greater proportion of the page so it is easier to read.

Thank you for your suggestion, we have increased the font size of Figure 2.

On p7 line 217-218, the authors are over-conservative in claiming “biological plausibility.” The literature, as they know (and use in their bibliography) is rich with repeated confirmations of the associations of positive life style factors and healthy behaviors with subsequent cardiac, metabolic morbidities, mortality from cardiometabolic disease, and all-cause mortality. In some instances RCTs are available, and in some specific cases (eg triglyceride-rich lipoproteins), there is support from Mendelian randomization studies.

We totally agree with you. We have changed this sentence by “This finding is consistent with multiple previous studies supporting the fact that following healthy habits results in better health.”

Pg 8 line 230—the authors are to be commended on this statement about HLS as potential health promotion tools. Perhaps another few sentences could be added (if they do not plan another publication about this application).

Thank you for yours comments, we have added: “Future research should be conducted in different scenarios, to better analyze the potential effects of HLS on healthy behaviors and health outcomes”.

The discussion on p 8-9 is masterful.

The authors are to be congratulated on their fine work and contribution. Thanks again.

Reviewer 2 Report

1.   Abstract: The age of participants should be included.

2. The introduction does not provide a clear rationale for the current study. The authors make the bold claim that healthy lifestyle habits associated with a lower risk of developing MetSyn and vice versa, with only few references are provided. None of these studies is addressed thoroughly. Are they cross-sectional or longitudinal? Are they conducted specifically with younger or older age groups? The clinical criteria for the diagnosis of metabolic syndrome in these studies are vague. Also which specific criteria for MetS will be considered in this study? A clear hypothesis should also be included at the close of the introduction. In general, this section is very weak and should be revised for clarity.

3.  Line 69-73: This paragraph should be enlarged.

4.  It is not clear to me how the study outcomes collected? I mean the clinical measurements of MetS.

5. Line 145: Clear rationale why participants were grouped into five categories?

6. It is totally unclear how the authors examined the relationship between healthy lifestyle habits at baseline and MetS after 6 years of follow-up. Please clarify if these two time points were treated as two points in the analyses.

7. In the analysis section, 11 covariates are mentioned which were used to adjust. Please clarify why these covariates were selected? I expect a strong collinearity and therefore some caution should be taken when all of them are used simultaneously. It is not clear to the reader why the authors are using a logistic regression analysis. Also, longitudinal analysis is not presented- The authors should clarify how each of the time point of data collection is used and what conclusions can be drawn from the analysis.

8.  The results make no sense, and tables 3 & 4 are totally unclear that should be revised to meet the study objectives. The relationship between healthy lifestyle habits and MetS is vague. The two time points should be clarified both in text and tables.

9. In general, the discussion section is disorganized and incoherent. The discussion for the outcome variables is underdeveloped/dissatisfying. There is little or no discussion regarding the longitudinal associations between healthy lifestyle habits and MetS. This point need to be clarified. Some findings should be better discussed (Line 241-243; Line 249-252; Line 265-268).

10. Line 216-218: This statement should be supported with appropriate references.

11. Line 219-220: Previous studies…… [26]. The authors referred to only one study here.

12. Line 220-229: Please clarify if these studies are cross-sectional or longitudinal.

13. Is it necessary to present the findings of other studies in the discussion?

14. In conclusion section, the authors don't mention the implications of this study.

Author Response

Thanks for your review. We request your questions point by point below. The corrections suggested by you and other reviewers are highlighted in the original manuscript in red print.

1.   Abstract: The age of participants should be included.

We have included age and gender data in the abstract

2. The introduction does not provide a clear rationale for the current study. The authors make the bold claim that healthy lifestyle habits associated with a lower risk of developing MetSyn and vice versa, with only few references are provided. None of these studies is addressed thoroughly. Are they cross-sectional or longitudinal? Are they conducted specifically with younger or older age groups? The clinical criteria for the diagnosis of metabolic syndrome in these studies are vague. Also which specific criteria for MetS will be considered in this study?  A clear hypothesis should also be included at the close of the introduction. In general, this section is very weak and should be revised for clarity.

Introduction has been modified, according to your suggestions. The studies referred have been better described, the MS criteria that we have used has been included. We have used theMS definition according to the IDF, AHA and National Heart, Lung, and Blood Institute harmonizing definition. Also a better hypothesis has been added.

3.  Line 69-73: This paragraph should be enlarged.

Paragraph has been modified and enlarged as suggested.

4.  It is not clear to me how the study outcomes collected? I mean the clinical measurements of MetS. 

A better explanation has been added on the reviewed version of the manuscript. 

5. Line 145: Clear rationale why participants were grouped into five categories?

This point is commented in line 128-130. We have reinforced this sentence  

6. It is totally unclear how the authors examined the relationship between healthy lifestyle habits at baseline and MetS after 6 years of follow-up. Please clarify if these two time points were treated as two points in the analyses.

The present study is a prospective cohort study. Data from the main independent variable (HLS) were collected in the baseline questionnaire (Q-0). The outcome assessment (MetSyn) were collected in the questionnaires at 6 and 8 years of follow-up (Q-6 and Q-8)

We have tried to clarify this point better in Materials and Methods

7. In the analysis section, 11 covariates are mentioned which were used to adjust. Please clarify why these covariates were selected? I expect a strong collinearity and therefore some caution should be taken when all of them are used simultaneously. It is not clear to the reader why the authors are using a logistic regression analysis. Also, longitudinal analysis is not presented- The authors should clarify how each of the time point of data collection is used and what conclusions can be drawn from the analysis.

We have tried to clarify this point by generating a new sub-section "Evaluation of other variables".

8.  The results make no sense, and tables 3 & 4 are totally unclear that should be revised to meet the study objectives. The relationshipbetween healthy lifestyle habits and MetS is vague. The two time points should be clarified both in text and tables.

There is a time line of around 6 years between the collection of data for the calculation of HLS and the measures reported for the diagnosis of MetSyn. We have reinforced this explanation in the tables and in the text

9. In general, the discussion section is disorganized and incoherent.The discussion for the outcome variables is underdeveloped/dissatisfying. There is little or no discussion regarding the longitudinal associations between healthy lifestyle habits and MetS. This point need to be clarified. Some findings should be better discussed (Line 241-243; Line 

We have tried to clarify this point in the discussion. We think the new discussion is more organized and coherent than before

10. Line 216-218: This statement should be supported with appropriate references.

This point has been corrected

11. Line 219-220: Previous studies…… [26]. The authors referred to only one study here.

We have added another reference

12. Line 220-229: Please clarify if these studies are cross-sectional or longitudinal. 

This point has been clarified

13. Is it necessary to present the findings of other studies in the discussion?

We have done changes in the discussion. 

14. In conclusion section, the authors don't mention the implications of this study.

Implications have been added in conclusion

Reviewer 3 Report

In this study, entitled “Healthy Lifestyle and incidence of metabolic syndrome in the SUN cohort”, Maria Garralda-Del-Villar et al. reported the significance of HLS on metabolic syndrome risk. Though the inverse association between various healthy life style parameters and MS prevalence is well studied previously, combinatorial effect of 9 HLS parameters on MS risk and huge sample size makes this study an important one in this field. I have some important comments about this study that may make this article much more significant.

Comments:

1.      Though exactly not similar to this study in terms of number and ethnicity of the subjects, similar studies were reported recently describing the association between HLS and MS prevalence (PMID: 25948783, 27831954). Authors need to include these studies in their introduction and explain how the current study is different from these studies.  

2.      In Figure 1, the number of subjects with energy intakes outside the defined values (n=1399) to be mentioned in the flow chart.

3.      In table 2, statistical significance values to incorporated and a dedicated paragraph to be incorporated in the results section that describes about the Table 2.    

4.      Authors reported that the data was not shown about the stratified analysis involving gender and age. Authors are suggested to submit this data as supplemental tables/figures.   

5.      Statistical significance values to be incorporated in the table 4.

6.      Figure 2, font size to be increased as reviewers are finding difficultly to read text in the figure.     

7.      References 15 and 18 were published in Spanish language. Authors need to mention whether these articles are available in English. If so, please report as like the authors did with reference 9.   

 Author Response

Thanks for your review. Your comments help us and encourage us to improve our research.

We request your questions point by point below. The corrections suggested by you and other reviewers are highlighted in the original manuscript in red print.

In this study, entitled “Healthy Lifestyle and incidence of metabolic syndrome in the SUN cohort”, Maria Garralda-Del-Villar et al. reported the significance of HLS on metabolic syndrome risk. Though the inverse association between various healthy life style parameters and MS prevalence is well studied previously, combinatorial effect of 9 HLS parameters on MS risk and huge sample size makes this study an important one in this field. I have some important comments about this study that may make this article much more significant. 

Comments:

1.      Though exactly not similar to this study in terms of number and ethnicity of the subjects, similar studies were reported recently describing the association between HLS and MS prevalence (PMID: 25948783, 27831954). Authors need to include these studies in their introduction and explain how the current study is different from these studies.  

Thanks for contributing these interesting references. We have included them in our introduction.

2.      In Figure 1, the number of subjects with energy intakes outside the defined values (n=1399) to be mentioned in the flow chart.

Thank you. The table was moved. Now is correct.

3.      In table 2, statistical significance values to incorporated and a dedicated paragraph to be incorporated in the results section that describes about the Table 2.    

We have incorporated a column with statistical significance and we have written more about this table

4.      Authors reported that the data was not shown about the stratified analysis involving gender and age. Authors are suggested to submit this data as supplemental tables/figures.   

We have done a supplemental figure.

5.      Statistical significance values to be incorporated in the table 4. 

We have incorporated p for trend in table 4. And we have changed the reference text in Result

6.      Figure 2, font size to be increased as reviewers are finding difficultly to read text in the figure.     

Thank you for your suggestion, we have increased the font size of Figure 2.

7.      References 15 and 18 were published in Spanish language. Authors need to mention whether these articles are available in English. If so, please report as like the authors did with reference 9.   

We have done changes in  these references

Round  2

Reviewer 2 Report

1. The age group of these studies is still unclear in the introduction. Are they conducted with younger or older age groups? (Line 55-73) Also, the clinical criteria for the diagnosis of metabolic syndrome in these studies are unclear to me (Line 61-62, Line 65); e.g. hyperinsulinemia, impaired fasting glucose, dyslipidemia…..etc.

2. Line 73: Please delete this repetitive sentence. This question is an…..etc.  

3. There is still no description of the clinical measurements of MetS. For example, serum triglycerides were measured with a Hitachi 704 Analyzer. HDL was measured with a……etc.

4.  It is still unclear to me why 11 covariates were chosen. The authors must provide a clear rationale why each covariate selected. For example, it is strange that the prevalence of cancer is chosen as a covariate.

Author Response

We highly appreciate this second review and comments, which help us to clarify more the research. 

1. The age group of these studies is still unclear in the introduction. Are they conducted with younger or older age groups? (Line 55-73). Also, the clinical criteria for the diagnosis of metabolic syndrome in these studies are unclear to me (Line 61-62, Line 65); e.g. hyperinsulinemia, impaired fasting glucose, dyslipidemia…etc.

We have made corrections on this matter; included the age groups, as well as, the criteria used to define Metabolic Syndrome in each article. We have tried to make it as clear as possible. Thank you for the recommendation.

2. Line 73: Please delete this repetitive sentence. This question is an…etc.  

We have erased this sentence and checked for any other similar mistakes.

3. There is still no description of the clinical measurements of MetS. For example, serum triglycerides were measured with a Hitachi 704 Analyzer. HDL was measured with a.…etc.

We have introduced the following explanation to clarify this information: 

“All the analytical parameters used in the validation of self-reported metabolic syndrome components were obtained from the Clinical Analyses Service of Clínica Universidad de Navarra (CUN). The analyses of glucose, HDL-cholesterol, total cholesterol and triglycerides were measured in blood serum with the analyzer equipment Roche/Hitachi Modular Analytics, through spectrophotometry by the enzymatic-colorimetric method with glucose oxidase and p-aminophenazone (GOD-PAP)”

Thank you for this suggestion we did not noticed.

4.  It is still unclear to me why 11 covariates were chosen. The authors must provide a clear rationale why each covariate selected. For example, it is strange that the prevalence of cancer is chosen as a covariate.

We have added this explanation: 

Our approach was to use the consideration of a priori causal knowledge to suggest which were the most relevant variables to be adjusted for. Causal diagrams were used to encode qualitative a priori subject matter knowledge. We did not use merely statistical criteria, because this statistics-only approach has been discouraged[29–31]. For instance, in reference to prevalence of cancer: it may lead to weight loss and, therefore, be associated with MetSyn. Besides, it can be related to a change in diet and lifestyle, and these can influence MetSyn.”

We have sent separably  a diagram which we hope it clarifies it more. 
